# *Caenorhabditis elegans* LET-381 and DMD-4 control development of the mesodermal HMC endothelial cell

Nikolaos Stefanakis§, Jasmine Xi*, Jessica Jiang‡ and Shai Shaham§

## ABSTRACT

Endothelial cells form the inner layer of blood vessels and play key roles in circulatory system development and function. A variety of endothelial cell types have been described through gene expression and transcriptome studies; nonetheless, the transcriptional programs that specify endothelial cell fate and maintenance are not well understood. To uncover such regulatory programs, we studied the *C. elegans* head mesodermal cell (HMC), a non-contractile mesodermal cell bearing molecular and functional similarities to vertebrate endothelial cells. Here, we demonstrate that a Forkhead transcription factor, LET-381, is required for HMC fate specification and maintenance of HMC gene expression. DMD-4, a DMRT transcription factor, acts downstream of and in conjunction with LET-381 to mediate these functions. Independently of LET-381, DMD-4 also represses the expression of genes associated with a different, non-HMC, mesodermal fate. Our studies uncover essential roles for FoxF transcriptional regulators in endothelial cell development and suggest that FoxF co-functioning target transcription factors promote specific non-contractile mesodermal fates.

KEY WORDS: Endothelial cell development, FoxF, *let-381*, DMRT, *dmd-4*, *Caenorhabditis elegans*

## INTRODUCTION

Endothelial cells are mesodermal cells that line blood vessels. They are crucial for maintaining circulatory system integrity, regulating blood flow and maintaining ion homeostasis of surrounding tissues. Specialized capillary endothelial cells are key components of the vertebrate blood-brain barrier. Aberrant endothelial cell development and function is associated with various diseases including cancer, atherosclerosis and stroke (Rohlenova et al., 2018).

Endothelial cells originate from the embryonic mesoderm germ layer. During vasculogenesis, mesodermal precursors of both endothelial and hematopoietic cells aggregate into cell clumps called blood islands. Later, the outer layer of these clumps differentiates into endothelial cells, while inner cells give rise to hematopoietic cells. During subsequent angiogenesis, the endothelial cell population expands and remodels, sprouting and branching to form the vascular lining (Marziano et al., 2021). Although all vasculature beds exhibit common features, endothelial cells are phenotypically heterogeneous, functionally tailored to the specific needs of the tissue in which they reside (Hennigs et al., 2021). Many studies have defined key roles for vascular endothelial growth factor (VEGF) and Notch signaling in the early steps of vascular development (Olsson et al., 2006; Phng and Gerhardt, 2009). ETS factors are also crucial for the specification and maintenance of endothelial cell fate (Gomez-Salinero et al., 2022; Marcelo et al., 2013). Furthermore, recent advances in multiomics technologies have facilitated molecular characterization of phenotypically-distinct endothelial cells, highlighting differences in gene expression and defining a host of endothelial subtypes (Trimm and Red-Horse, 2023). Nonetheless, the transcriptional programs involved in endothelial cell differentiation and subtype specification and maintenance remain incompletely understood.

The nematode *Caenorhabditis elegans* has played pivotal roles in the identification of conserved molecular mechanisms of development and function of different tissues and cell types (Ambros, 2011; Gieseler et al., 2017; Horowitz and Shaham, 2024; Kratsios and Hobert, 2024; Richmond, 2005; Singhvi et al., 2024). The *C. elegans* head mesodermal cell (HMC) was recently shown to resemble vertebrate endothelia (Choi et al., 2023). Like endothelial cells, the HMC is a non-contractile, mesodermal cell. It resides within the animal's main body cavity, the pseudocoelom, a simple circulatory system that provides a means for nutrient, oxygen and signaling molecule distribution (Hall and Altun, 2008). G-protein-coupled receptors (GPCR) on the HMC respond to peptidergic signals to elicit changes in HMC intracellular calcium levels, leading to contraction of surrounding muscles connected to the HMC via gap junctions (Choi et al., 2023; Hall and Altun, 2008). Similarly, vertebrate endothelial cells can change their intracellular calcium levels in response to extracellular peptides and control smooth muscle contraction via myoendothelial gap junctions (Figueroa and Duling, 2009; Griffith, 2004; Maguire and Davenport, 2005).

To gain insights into the molecular mechanisms controlling specification and maintenance of endothelial cell fates, we therefore investigated HMC development. Anatomically, the HMC cell body lies dorsomedially, just above the posterior pharyngeal bulb (Fig. 1A). It extends a short anterior and a long dorsal posterior process as well as two lateral processes that project around the pharynx and merge ventrally, where they extend anteriorly and posteriorly to mirror the dorsal anterior/posterior processes. We found that LET-381, a FoxF transcription factor continuously expressed in the HMC, is required for both the initial specification and subsequent maintenance of HMC identity. The LET-381 target DMD-4, a DMRT transcription factor, acts with LET-381 to specify HMC fate and to maintain HMC gene expression. DMD-4 also represses expression of genes normally expressed in GLR glia, a different mesodermal cell type. Previous studies have shown that

Laboratory of Developmental Genetics, The Rockefeller University, 1230 York Avenue, New York, NY 10065, USA.
*Present address: Brown University, 69 Brown St., Providence, RI 02912, USA.
‡Present address: University of Pittsburg School of Medicine, Pittsburgh, PA 15261, USA.

§Authors for correspondence (nikosistefanakis@gmail.com; shaham@rockefeller.edu)

N.S., 0000-0003-0921-987X; S.S., 0000-0002-3751-975X

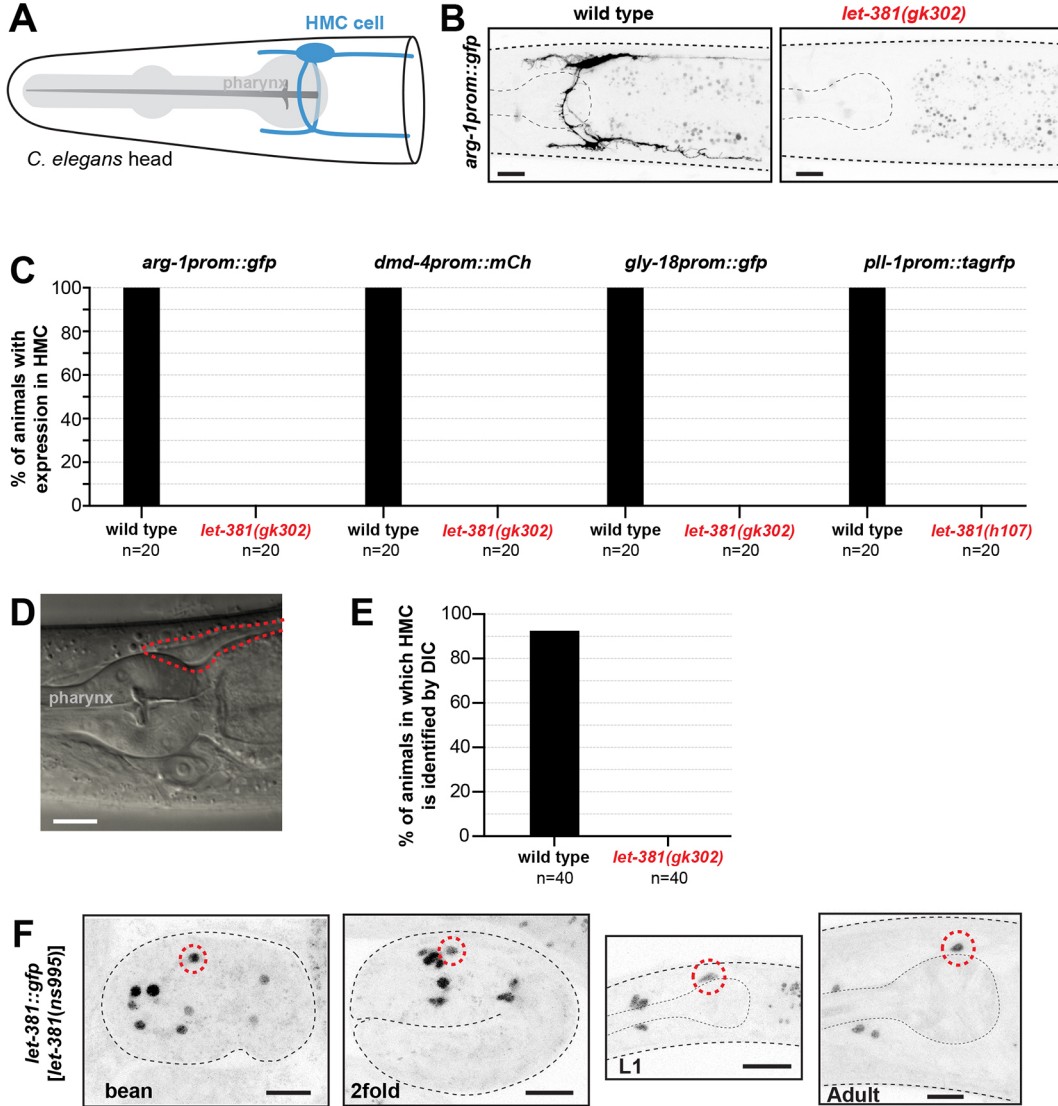

**Fig. 1. *let-381* is required for HMC specification.** (A) Schematic representation of the HMC cell (blue). (B) Fluorescence images of the HMC-specific *arg-1prom:: gfp* reporter in wild type (left) and *let-381(gk302)* mutant (right). Expression is not observed in the mutant. (C) Percentage of animals with expression in HMC of four different reporters (indicated above the bars) for wild type and *let-381* mutants. Expression in the HMC is not detected in the mutants. (D) DIC image of young adult wild-type animal posterior head region (see Fig. 1A). (E) Percentage of young adult animals in which the HMC cell body can be identified by DIC microscopy in wild type and *let-381(gk302)* mutants. (F) Expression of *let-381::gfp* in embryonic stages (bean, 2fold), L1 larva and adult animals. *n*=20 animals for each genotype in C and *n*=40 for each genotype in E. Anterior is left, dorsal is up. Red dashed lines indicate the HMC cell body in D or HMC nucleus in F. Gray dashed lines outline the animal and the pharynx. Scale bars: 10 μm.

LET-381 (FoxF) acts with UNC-30 (Pitx) and CEH-34 (Six2) to control the development of GLR glia and mesodermally-derived coelomocytes, respectively. Taken together, our studies reveal that LET-381 acts as a terminal selector factor for endothelial cell differentiation and demonstrate that its co-functional target transcription factors specify the identities, and control maintenance, of distinct mesodermal fates.

## RESULTS

### *let-381* is required for HMC cell specification

We previously reported that the LET-381 transcriptional regulator is expressed in GLR glia, coelomocytes and the HMC (Amin et al., 2010; Stefanakis et al., 2024). While roles for LET-381 in GLR and coelomocyte development are well established (Amin et al., 2010; Stefanakis et al., 2024), whether LET-381 also promotes HMC specification was unknown. To address this

question, we introduced HMC reporter transgenes (*arg-1prom:: gfp*, *dmd-4prom::mCherry*, *gly-18prom::gfp*, *pll-1prom1::rfp*) into animals homozygous for the *let-381(gk302)* or *let-381(h107)* alleles. *let-381(gk302)* animals contain a deletion removing LET-381 DNA binding domain-encoding sequences and *let-381(h107)* animals harbor a splice acceptor point mutation predicted to result in a truncated LET-381 protein (Barstead et al., 2012; Howell et al., 1987). While animals homozygous for *gk302* and *h107* undergo late-embryonic or early-larval developmental arrest, respectively, some escapers develop further to become sterile adults. We found that neither *let-381(h107)* arrested larvae nor *let-381(gk302)* adult escapers expressed any of the HMC reporters we tested (Fig. 1B,C). Furthermore, while the HMC cell body was easily identified by its characteristic shape and position in young adult wild-type animals using differential interference contrast (DIC) microscopy, no HMC cell body was discerned in young adult escaper *let-381(gk302)*

mutants (Fig. 1D,E). Taken together, these experiments suggest that LET-381 is required for HMC specification.

### let-381 is continuously and cell-autonomously required to maintain HMC identity

To determine when and where LET-381 is required to promote HMC specification, we tracked expression of a CRISPR/Cas9-generated *let-381::gfp* reporter in which *gfp* coding sequences are inserted into the endogenous *let-381* locus. As shown in Fig. 1F, *let-381::gfp* expression in the HMC cell begins in bean-stage embryos and is maintained throughout adulthood, suggesting that LET-381 may be required not only to specify the HMC, but also to maintain its fate.

To test this idea, we sought to directly determine whether LET-381 is required for HMC fate maintenance. We have previously shown that LET-381 promotes its own expression in GLR glia using an autoregulatory *let-381* binding motif located upstream of the first exon. A deletion/insertion genomic lesion removing this motif,

*let-381(ns1026)*, does not affect LET-381 embryonic GLR glia expression, but blocks continued expression in these cells beyond the first larval stage (Stefanakis et al., 2024). While the *let-381(ns1026)* mutation did not affect LET-381 expression in the HMC, another allele we generated, *let-381(ns1023)*, lacking sequences surrounding the autoregulatory *let-381* binding motif, preserved embryonic expression but blocked postembryonic expression of LET-381 in both the HMC and GLR glia (Fig. 2A-C). A possible explanation for this difference is that the 34 bp insertion present in *let-381(ns1026)*, but not in *let-381(ns1023)*, serendipitously binds an HMC-specific transcription factor allowing LET-381 expression. Importantly, we found that expression of six known HMC markers (*pll-1::gfp*, *gbb-2::gfp*, *snf-11<sup>fosmid</sup>::mCherry*, *gly-18prom::gfp*, *hot-2prom::gfp*, *glb-26prom::gfp*) gradually waned and was nearly abolished by the L4 larval stage in *let-381(ns1023)* autoregulatory mutants (Fig. 3A,B; Fig. S1A-D). Expression levels of a *dmd-4prom::mCherry* HMC reporter were likewise substantially reduced (Fig. 3C,D). By contrast, expression of these reporters in the HMC was not affected in the GLR

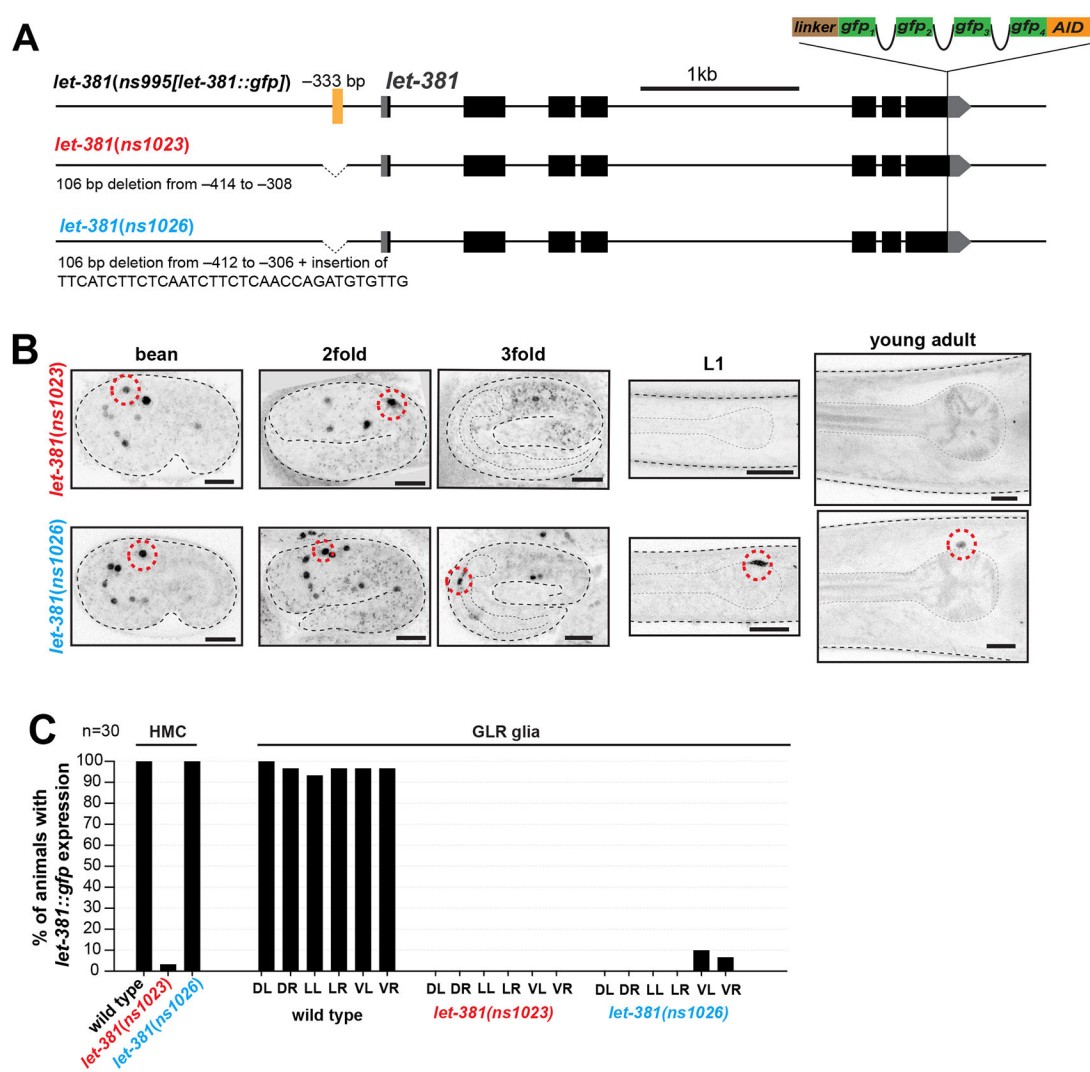

**Fig. 2. A *let-381* autoregulatory motif is required to maintain LET-381 expression in the HMC.** (A) Schematics showing details of *let-381* autoregulatory motif deletion mutant alleles. Both alleles remove a sequence containing the *let-381* autoregulatory motif (yellow line), located at −333 bp from the ATG, but *let-381(ns1026)* also has an insertion. (B) Images of *let-381::gfp* shown for bean, 2-fold and 3-fold embryonic stages, L1 larva and young adults for each genotype. The *let-381(ns1023)* allele (upper panel row) affects maintenance of *let-381::gfp* expression in both HMC and GLR glia. *let-381(ns1026)* (lower panel row) affects maintenance only in GLR glia. (C) Percentage of young adult wild-type, *let-381(ns1023)* and *let-381(ns1026)* animals with *let-381::gfp* expression in HMC and in each of the six GLR glia. *let-381::gfp* expression is nearly abolished from both the HMC and GLR glia in *let-381(ns1023)*. *n*=30 animals for each genotype scored in C. Anterior is left, dorsal is up. Red dashed lines indicate the HMC nucleus. Gray dashed lines outline the animal and the pharynx. Scale bars: 10 μm.

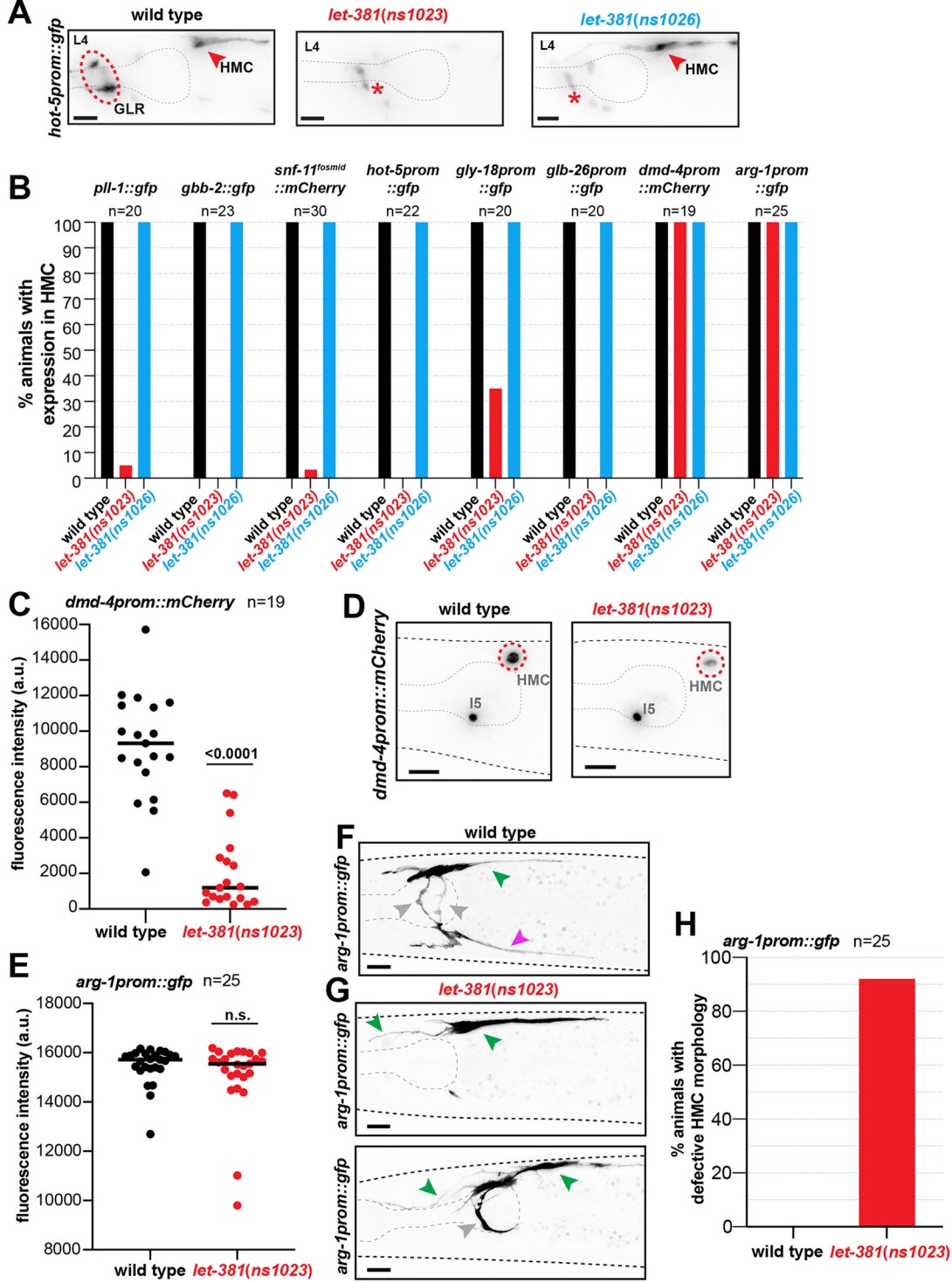

**Fig. 3. LET-381 is required for maintenance of HMC expression and morphology.** (A) *hot-5prom::gfp* is expressed in both HMC (red arrowhead) and GLR glia (red dashed circle) in wild-type animals. Expression of this reporter is lost in both cell types in *let-381*(*ns1023*) animals. By contrast, expression is lost only GLR glia in *let-381*(*ns1026*) animals. Red asterisk shows unrelated expression of *hot-5prom::gfp* in neurons. (B) Percentage of animals with expression in HMC of eight different reporters (indicated above the bars) for wild type, *let-381*(*ns1023*) and *let-381*(*ns1026*). (C,D) Quantification of fluorescence intensity of *dmd-4prom::mCherry* reporter in HMC in wild-type and *let-381*(*ns1023*) animals (C). *dmd-4prom::mCherry* expression is significantly reduced in *let-381*(*ns1023*). Representative images for each genotype shown in D. (E-G) Quantification of fluorescence intensity of the *arg-1prom::gfp* reporter in wild-type and *ns1023* animals. Fluorescence intensity in the cell body is not affected in *let-381*(*ns1023*) (E). Representative images for each genotype shown in F and G. Ventral process (magenta arrowhead) is missing, dorsal process (green arrowheads) is anteriorly mis-extended and lateral processes (gray arrowhead) are missing (top) or mis-extended (bottom) in *let-381(ns1023)* animals (G). (H) Percentage of animals with HMC morphology defects in wild type and *let-381(ns1023)* mutants, as assessed with the *arg-1prom::gfp* reporter. Number of animals (*n*) scored for each genotype for each reporter is shown under or next to reporter transgene name in B, C, E and G. Black lines on dot-plots indicate mean (C,E). Unpaired *t*-test was used for statistical analyses in C and E. n.s., not significant. a.u., arbitrary units. Anterior is left, dorsal is up. Gray dashed lines outline the animal and the pharynx. Scale bars: 10 μm.

glia-specific *let-381(ns1026)* autoregulatory mutant (Fig. 3A,B). These observations support the conclusion that *let-381* acts cell autonomously to maintain HMC gene expression.

To assess whether LET-381 is required to maintain HMC morphology, we took advantage of our finding that HMC expression of *arg-1prom::gfp* was not affected in *let-381(ns1023)* animals (Fig. 3B,E), allowing us to visualize HMC cell shape. We found that in *let-381(ns1023)* animals, the dorsal anterior HMC process was often misextended, while lateral processes could be shorter, bifurcated, misextended towards the anterior of the animal or missing altogether. Occasionally, the lateral processes extended ventrally, but failed to extend a ventral process. In total, 92% of *arg-1prom::gfp; let-381(ns1023)* animals displayed HMC cell shape defects (Fig. 3F-H; Movies 1-4). Thus, LET-381 is continuously required in the HMC to maintain gene expression and cell morphology.

### *dmd-4* is also required for HMC fate and gene expression
Our previous studies have demonstrated that LET-381 acts in conjunction with one of its targets, the transcription factor UNC-30, to specify GLR glia fate (Stefanakis et al., 2024). Similarly, another

study has revealed that a LET-381-regulated transcription factor, CEH-34, acts with LET-381 to drive post-embryonic coelomocyte fate acquisition (Amin et al., 2010). We wondered, therefore, whether a similar regulatory scheme drives HMC development. The Doublesex Mab3-related transcription factor DMD-4 (DMRT) is required for gene expression in the HMC (Bayer et al., 2020; Stefanakis et al., 2024) and its own expression was regulated in the HMC by LET-381 (Fig. 3C,D). Thus, like UNC-30 and CEH-34 in GLR glia and coelomocytes, respectively, DMD-4 might function together with LET-381 for HMC fate specification. To test this idea, we examined expression of five additional HMC gene reporters in *dmd-4(ot933)* mutants lacking sequences encoding the DMD-4 DNA binding domain. While most *dmd-4(ot933)* mutants die as embryos, some escape lethality and reach adulthood (Bayer et al., 2020). We found that none of the five HMC reporters we tested, including *let-381::gfp*, was expressed in these escapers (Fig. 4A,B). Furthermore, as in *let-381* mutants, the HMC cell body was also not detected by DIC microscopy in young adult *dmd-4(ot933)* mutants (Fig. 4C). These results suggest that the HMC is not specified in the absence of *dmd-4*. Indeed, *dmd-4(ot933)* animals exhibited

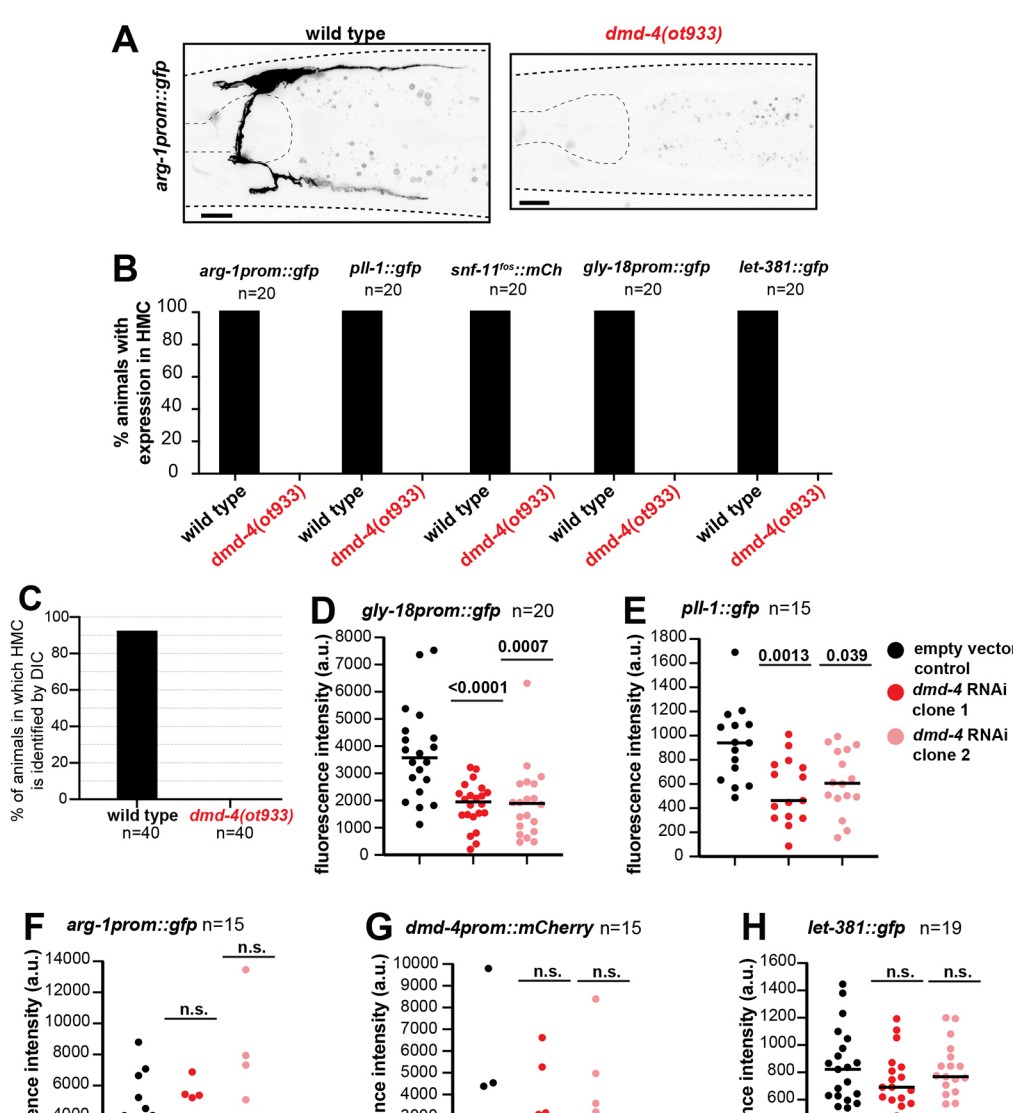

**Fig. 4. DMD-4 is required for HMC fate specification and HMC gene expression.** (A) HMC-specific *arg-1prom*::*gfp* reporter expression in wild-type (left) and *dmd-4(ot933)* animals (right). Expression is not observed in *dmd-4(ot933)*. (B) Percentage of animals with expression in HMC of five different reporters (indicated above the bars) for wild type and *dmd-4(ot933)*. Expression in the HMC is not detected in the *ot933* mutants for all five reporters. (C) Percentage of young adult animals in which the HMC cell body can be identified by DIC microscopy in wild type and *dmd-4(ot933)* mutants. (D-H) Quantification of fluorescence intensity for *gly-18prom*::*gfp* (D), *pll-1*::*gfp* (E), *arg-1prom*::*gfp* (F), *dmd-4prom*::*mCherry* (G) and *let-381*::*gfp* (H) reporters in control and *dmd-4* RNAi animals. Number of animals (*n*) scored for each genotype for each reporter is shown under or next to reporter transgene name in B, D-I and under the genotype for C. Black lines on dot-plots indicate mean (D-H). Unpaired *t*-test was used for statistical analysis in D-H. n.s., not significant. a.u., arbitrary units. Anterior is left, dorsal is up. Gray dashed lines outline the animal and the pharynx. Scale bars: 10 µm.

accumulation of food material in their anterior intestine (Bayer et al., 2020), consistent with a previously-described role for the HMC in controlling the *C. elegans* digestive cycle (Choi et al., 2023).

To distinguish between fate specification and maintenance roles of DMD-4, we examined the expression of HMC reporters in animals grown postembryonically for 3 days on bacteria expressing dsRNA against *dmd-4*. We found that induction of RNAi using two different bacterial dsRNA vectors, targeting either a portion of or the entire *dmd-4* mRNA (*dmd-4* RNAi clone 1 and clone 2, respectively), downregulated expression of the *gly-18prom::gfp* and *pll-1::gfp* HMC reporters, but not expression of *arg-1prom::gfp* (Fig. 4D-F). Thus, DMD-4 is required for maintaining expression of some HMC genes. Expression of a *dmd-4prom::mCherry* transcriptional reporter was not affected by *dmd-4*(RNAi), suggesting that DMD-4 is not required to maintain its own expression in HMC (Fig. 4G).

Importantly, *let-381::gfp* expression, which was affected in *dmd-4*(*ot933*) mutants, was not affected by *dmd-4*(RNAi) (Fig. 4H), suggesting that, as with UNC-30 and CEH-34, DMD-4 acts downstream of LET-381 in maintenance of the HMC fate. We conclude, therefore, that DMD-4 acts together with LET-381 in HMC fate specification and maintenance.

### LET-381 and DMD-4 binding motifs are required for GLR glia gene expression

To determine whether LET-381 and DMD-4 co-regulate gene expression in the HMC, we scanned non-exonic regions of the LET-381 target genes in Fig. 3B for LET-381 binding motifs (TGTTTABA; Stefanakis et al., 2024) and used CRISPR/Cas9 to mutate these sites. Mutating *let-381* motifs upstream of the genes *pll-1* or *gbb-2*, endogenously fused to *gfp*, abolished *gfp* expression in HMC (Fig. 5A-C). Similarly, deleting two predicted *dmd-4* sites (NWGTAWCNNN; Narasimhan et al., 2015) upstream of *pll-1*, whose expression was downregulated in DMD-4 RNAi (Fig. 4E), significantly reduced *pll-1::gfp* expression (Fig. 5D). We conclude, therefore, that LET-381 and DMD-4 likely control HMC gene expression through binding to at least some common target genes.

### LET-381 and DMD-4 are not sufficient to induce HMC gene expression

The broad effects of *let-381* and *dmd-4* on HMC gene expression suggests that these transcription factors could be sufficient to induce HMC gene expression when expressed in naïve cells. However, we found that inducible misexpression, using a heat-shock promoter, of either or both genes did not result in misexpression of the *glb-26prom::gfp* HMC reporter (Fig. S2A). Thus, LET-381 and DMD-4 are likely not sufficient to induce HMC fate alone, and additional transcription factors may be necessary to induce HMC fate.

### DMD-4 represses expression of some GLR glia genes in the HMC

We previously showed that UNC-30, which acts together with LET-381 in GLR glia, represses expression of HMC genes in these glia (Stefanakis et al., 2024). We wondered, therefore, whether DMD-4 reciprocally represses GLR glia gene expression in the HMC. Indeed, we found that ~50% of *dmd-4*(RNAi) animals expressed the GLR glia-specific marker *nep-2prom7::gfp* ectopically in the HMC cell (Fig. 6A,B). This ectopic expression appeared to be dimmer than *nep-2prom7::gfp* expression in GLR glia (Fig. 6B). Three other GLR glia reporters, with a lower GLR glia expression than *nep-2prom7::gfp*, did not show ectopic expression in the HMC upon *dmd-4*(RNAi) (Fig. S3A). To more

robustly knock down DMD-4 expression and in a temporally controlled manner, we used CRISPR/Cas12 to insert sequences encoding auxin inducible degrons (mIAA7) into the *dmd-4* genomic locus and assessed the dynamics of ectopic *nep-2p7::gfp* expression in HMC following auxin addition starting either at the L1 or L4 larval stage (Fig. 6C; Fig. S3B). In the presence of the auxin derivative 5-phenyl-1H-indole-3-acetic acid (5-Ph-IAA) and ubiquitously expressed $_{At}$TIR1$^{(F79G)}$ (Hills-Muckey et al., 2021; Negishi et al., 2021; Sepers et al., 2022), *DMD-4::linker::mIAA7::wrmScarletI3::mIAA7* tagged protein was rapidly degraded within 1 h (Fig. 6D; Fig. S3C). Ectopic *nep-2p7::gfp* was first observed in HMC cells after 16 h on 5-Ph-IAA for L1 treated animals and after 6 h on 5-Ph-IAA for L4 treated animals (Fig. 6E; Fig. S3D), at which time HMC morphology still appeared to be wild-type (Fig. 6F). After 24 h on 5-Ph-IAA, ~80% of animals expressed *nep-2p7::gfp* in the HMC (Fig. 6E), with some exhibiting abnormal HMC morphology (e.g. shorter, thinner processes). By 48-72 h on 5-Ph-IAA, all *nep-2p7::gfp*-expressing HMCs displayed significant morphological defects (Fig. 6G; Fig. S3E,F), and some HMCs appeared to be dying (big round cell; Fig. S3G). Furthermore, in contrast to *dmd-4*(RNAi), another GLR-reporter, *lgc-55prom::gfp* was also expressed in HMC cells upon DMD-4 AID (Fig. S3H-J). Together, these results suggest that DMD-4 acts to repress at least some GLR glia genes in the HMC and to maintain HMC morphology and cell survival.

### DISCUSSION

We describe a gene regulatory network governing fate specification and maintenance of the *C. elegans* endothelial-like HMC (Fig. 6H). Early in development, the LET-381 transcription factor specifies HMC fate, while later it is continuously required to maintain HMC gene expression and morphology. An autoregulatory sequence upstream of the *let-381* locus ensures continual LET-381 expression. DMD-4, another transcriptional regulator and a target of LET-381, acts with LET-381 for both HMC fate specification and maintenance. DMD-4, in turn, also represses GLR glia gene expression in the HMC.

*C. elegans* LET-381 is expressed in only three cell types: GLR glia, which approximate the inner aspect of the central neuropil, the nerve ring, and exhibit astrocyte and endothelial characteristics (Stefanakis et al., 2024); coelomocytes, liver-like detoxifying cells residing within the coelomic cavity (Fares and Greenwald, 2001; Hall and Altun, 2008); and the endothelial-like HMC. All three cell types are non-contractile cells that derive from the mesoderm-like lineage of the MS blast cell. This study, together with previously published results, shows that LET-381 acts as a master regulator that specifies and maintains all three cell types, and does so by co-regulating a wide variety of cell-type specific genes and cell morphologies (Stefanakis et al., 2024; Amin et al., 2010). How does a single regulator specify such different cell types? Our findings suggest that specificity is enabled via collaboration with co-functional target transcription factors. Specifically, DMD-4, UNC-30 and CEH-34 act with LET-381 to control development of the HMC, GLR glia and coelomocytes, respectively. A similar regulatory strategy is observed in the nervous system, where master regulatory factors, termed terminal selectors, combine with different co-acting transcriptional regulators to specify and maintain a variety of neuronal features, including gene expression and connectivity (Allan and Thor, 2015; Hobert and Kratsios, 2019; Leyva-Díaz et al., 2020). Indeed, assigning different co-acting transcription factors to a common core transcription factor to form core regulatory complexes (CoRCs), which direct expression of distinct effector

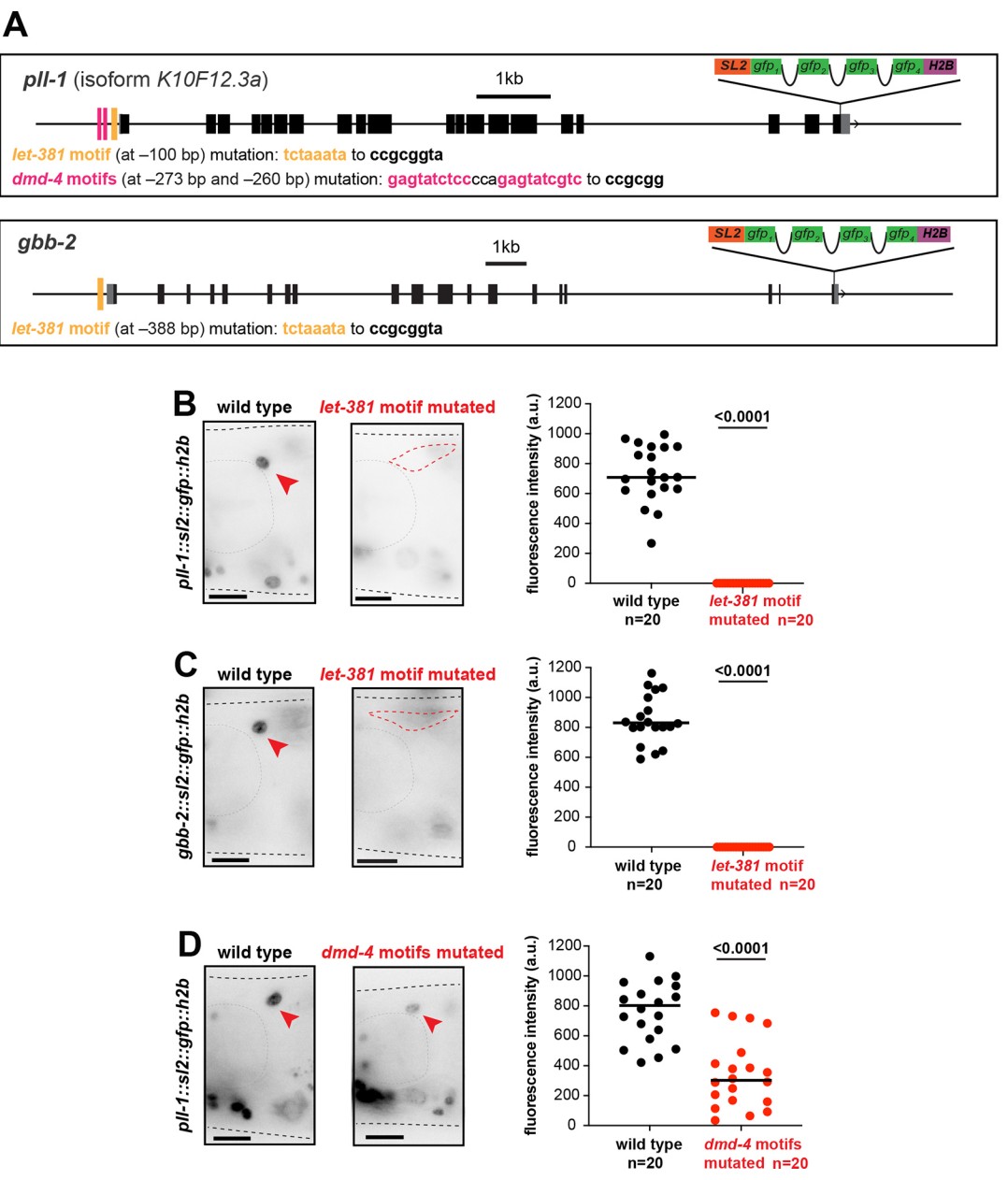

**Fig. 5. *let-381* and *dmd-4* motifs are required for endogenous gene expression in the HMC.** (A) Schematics of gene models, endogenous *gfp*-based tags and *let-381* motifs (yellow bars) and *dmd-4* motifs (magenta bars) for the genes *pll-1* and *gbb-2*. Distance from ATG and nucleotide changes for each motif mutation is shown below the gene models. (B,C) Expression of endogenously *gfp*-tagged *pll-1* (B) and *gbb-2* (C) in wild-type and *let-381* motif-mutated animals. (D) Expression of endogenously *gfp*-tagged *pll-1* in wild-type and *dmd-4* motif-mutated animals. For panels B-D, animal images are on the left, quantifications are shown in the dot-plots on the right. Red arrowhead indicates HMC; red dashed circle outlines HMC cell body in cases where HMC expression is not detected. Black lines on dot-plots indicate mean. Unpaired *t*-test used for statistical analysis. a.u., arbitrary units. Anterior is left, dorsal is up. Gray dashed lines outline the animal and the pharynx. Scale bars: 10 μm.

gene sets, is a widespread regulatory strategy that can be effectively used to generate novel cell types during evolution (Arendt et al., 2016).

Terminal selectors typically co-activate many cell-type-specific target genes. Recent findings suggest that in addition to gene activation, terminal selectors also repress gene expression of alternative cell types (Feng et al., 2020; Remesal et al., 2020; Reilly et al., 2022). Our findings suggest that such repression is mediated not by the CoRC core-factor, LET-381 here, but by its cofactors, DMD-4 and UNC-30, each repressing gene expression of the cell fate instructed by the other. It is thus possible that apart from

establishing cell-type specificity, terminal selector cofactors safeguard cell fate through repression of alternative fates.

Terminal selectors have also been implicated in controlling cell morphology, and our studies here are consistent with this idea, as *let-381* mutants exhibit alterations in HMC cell shape (Fig. 3F). Identification of specific morphology-related HMC target genes could, therefore, unveil molecular mechanisms controlling cell shape.

Although FoxF genes have conserved roles in visceral muscle development (Ormestad et al., 2006; Zaffran et al., 2001), roles for these genes in the specification of non-contractile cells are now also emerging. *foxf1* knockdown in planaria results in loss of several

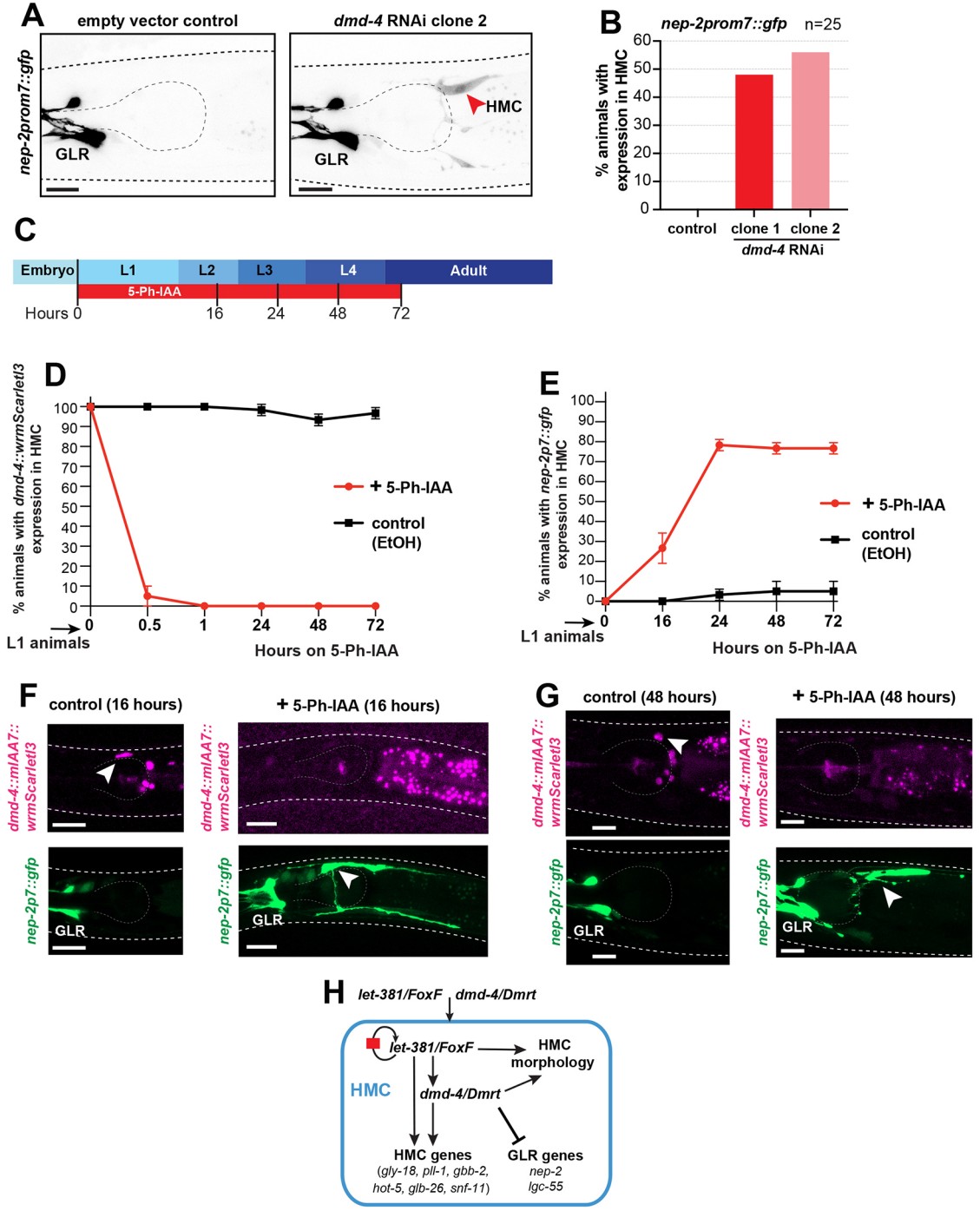

**Fig. 6. DMD-4 is required for repression of GLR glia genes in the HMC.** (A) Representative images showing expression of *nep-2prom7*::*gfp* in control and *dmd-4*(RNAi) animals. Red arrowhead indicates HMC. (B) Percentage of animals with ectopic HMC expression of the GLR-specific reporter *nep-2prom7*::*gfp* upon *dmd-4* RNAi. *n*=25 animals for each condition. (C) Timeline of DMD-4 auxin inducible degradation for L1 stage animals (L1 stage DMD-4 AID), showing time points relevant for D and E. (D) Quantification of DMD-4 degradation, based on wrmScarletI3 expression in HMC, at different time points for L1 stage DMD-4 AID. (E) Percentage of animals with ectopic HMC expression of the GLR-specific reporter *nep-2prom7*::*gfp* upon L1 stage DMD-4 AID at different time points. For D and E, three replicate experiments were performed, with *n*=20 animals per replicate for each condition. Error bars show standard deviation between replicates. (F,G) Representative images showing *dmd-4::mIAA7::wrmScarletI3::mIAA7* (magenta; top) and *nep-2p7*::*gfp* (green; bottom) for control and 5-Ph-IAA-treated animals at different times on 5-Ph-IAA. White arrowheads indicate HMC. HMCs with ectopic *nep-2p7*::*gfp* have wild-type appearance at 16 h on 5-Ph-IAA, but display defects at 48 h on 5-Ph-IAA. (H) Schematic representation summarizing the regulatory network for HMC development identified in this study. Anterior is left and dorsal is up for all images. Gray dashed lines outline the animal and the pharynx. Scale bars: 10 µm.

cell types, including phagocytic glia and pigment cells (Scimone et al., 2018). In mammals, *Foxf2* is required for differentiation of mesodermal pericytes (Reyahi et al., 2015). Inactivation of murine *Foxf1* results in complete absence of vasculogenesis in the yolk sac

(Mahlapuu et al., 2001). Although this defect appears to be due to an early role of *Foxf1* in splanchnic mesoderm and before endothelial cell specification (Astorga and Carlsson, 2007), single cell RNA-seq studies show that FoxF genes are enriched in certain

endothelial cells in the adult brain and lungs (Hupe et al., 2017; Paik et al., 2020; Wang et al., 2024; Kalucka et al., 2020). Functional roles for these genes in the endothelium have not been investigated. We believe it is possible, and perhaps likely, that as in *C. elegans*, FoxF factors cell autonomously specify and maintain mammalian endothelial cell fates and work together with specific CoRC cofactors to give rise to endothelial cell heterogeneity in different tissues. Indeed, Foxf2, expressed specifically in brain endothelial cells, is sufficient to induce expression of blood-brain-barrier-associated markers in cell culture (Hupe et al., 2017), and homologs of some LET-381 target genes we identified, including *gly-18/ GCNT2*, *gbb-2/GABBR2* and *pll-1/PLCL1*, are enriched in brain endothelia (Paik et al., 2020; Saunders et al., 2018). However, further investigation is required to clarify whether FoxF genes have such roles *in vivo*. Intriguingly, DMRT factors are expressed at low levels in brain endothelial cells, raising the possibility that, in addition to their conserved roles in sexual development, neurogenesis and muscle development (Hong et al., 2007), they may also have roles in endothelial cell specification.

## MATERIALS AND METHODS

### *C. elegans* strains and handling
Animals were grown on nematode growth media (NGM) plates seeded with *Escherichia coli* (OP50) bacteria as a food source unless otherwise mentioned. Strains were maintained by standard methods (Brenner, 1974). Wild type is strain N2, *C. elegans* variety Bristol (RRID:WB-STRAIN: WBStrain00000001). See Table S1 for a complete list of strains generated and used in this study. A few of the strains have been previously published and/or obtained from the *Caenorhabditis* Genetics Center (CGC).

### Genome engineering
Generation of endogenous deletions and motif mutations was performed using CRISPR/Cas9 tracrRNAs and crRNAs from Integrated DNA Technologies (IDT) as previously described (Dokshin et al., 2018). *let-381*(*ns1023*) and the previously described *let-381*(*ns1026*) deletion alleles were generated by use of two crRNAs (tggttgaagagacatacatc, ttatggatggaaaacagacg) and a single-stranded oligodeoxynucleotide (ssODN) (tcatcatacttttttccctctatcttctcaaccagatctgttttccatccataagccaccacccccattctgc). CRISPR/Cas9-generated different deletion alleles were: *ns1023* carrying a 106 bp deletion from −414 to −308 and *ns1026*, an indel carrying a deletion from −412 to −306 and insertion of a random 34 bp sequence (ttcatcttctcaatcttctcaaccagatgtgttg). Both alleles remove the tgtttata *let-381* motif at −333 bp from the ATG.

*pll-1*(*ns1040[*syb5792]*) is a substitution of −273 to −251, containing two consecutive *dmd-4* motifs, with ccgcgg, performed with crRNA (acgtcccagacgatactctg) and ssODN gcaatcccgcgcgccgcaggttccccaaccaaat-ccgcggtgggacgtccctttttgcttccgaaaaaataaaat. *pll-1*(*ns1040[*syb5792]*) is a substitution of *let-381* motif at −100 from tctaaata to ccgcggta, which was previously described in Stefanakis et al. (2024). *gbb-2*(*ns1043[*syb5759]*) is a mutation of *let-381* motif at −208 from TGTTTA to CCGCGG and a mutation of the PAM site at −228 from CC to AA, performed using crRNA (acaatcagcactaagaaaat) and ssODN taaaagttttcaaaaaaaaatataaataaataaataaa-attttcttagtgctgatccgcggtataatctcacacaacagctggcacccgcaatttg.

Generation of *dmd-4*(*ns1103[dmd-4::linker::mIAA7::::wrmScarletI3:: mIAA7]*) was performed with Cas12 using the protocol previously described (Ghanta and Mello, 2020). We used a single crRNA (agataaatttattatgacgat) and an ssODN repair donor template. The ssODN, including the desired insertion sequence flanked by homology arms (and mutation of the PAM site from caaa to tttt), was prepared by enzymatic digestion of PCR product into single-stranded DNA as described in Eroglu et al. (2023). Details of the inserted sequence, including the homology arms, are provided in Fig. S4.

### Generation of transgenic reporters
The *glb-26prom1::gfp* reporter was generated by a PCR fusion approach (Hobert, 2002). Genomic promoter fragments were fused to *gfp* followed by

the *unc-54* 3′ untranslated region (UTR). Promoters were initially amplified with primers A (gactgtgggagacgatcgtac) and B (ctctagagtcgacctgcaggcatg-caagctctgggaatgagcacacgaaa) from N2 genomic DNA. *gfp*, followed by *unc-54* 3′UTR, was amplified by primers C (agcttgcatgcctgcaggtcg) and D (aagggcccgtacggccgacta) from plasmid pPD9575. For the fusion step, PCR amplification was performed using primers A* (gttcgaagatctgcacgaag) and D* (caagaaaaacgccgtcctcg) as previously described (Hobert, 2002). PCR fusion DNA fragments were injected as simple extrachromosomal arrays in the wild-type N2 strain in the following concentrations: *glb-26prom1::gfp*, 50 ng/µl; *myo-3prom::mCherry* (co-injection marker), 25 ng/µl; pBluescript SK+, 25 ng/µl. The integrated array, *nsIs1052*, was generated by exposing animals to 33.4 µg/ml trioxsalen (Sigma-Aldrich, T6137) and UV irradiation using a Stratagene Stratalinker UV 2400 Crosslinker (360 µJ/cm²×100) as previously described (Kage-Nakadai et al., 2014).

### Generation of new *dmd-4* RNAi clone
A 4544 bp fragment containing the entire *dmd-4* genomic locus was PCR amplified from N2 genomic DNA with primers agaccggcagatctgatatcatc-gatgaattcgagctccagtcgagctccgcctacaatc and gcgcgtaatacgactcactatagggcg-aattgggtaccgggaggggatttgccacaagta and subsequently cloned by Gibson cloning (Gibson et al., 2009) into the empty RNAi vector L4440, which was PCR amplified by primers ccggtacccaattcgccta and tggagctcgaattcatcgat. Ligated plasmids were then transformed into HT115 *E. coli* bacteria by electroporation. Plasmid clones containing the properly inserted *dmd-4* locus were identified by whole plasmid sequencing, performed by Plasmidsaurus using Oxford Nanopore Technology with custom analysis and annotation.

### RNAi by feeding
Synchronized L1 larvae were placed on NGM plates with 1 mM IPTG and 25 µg/ml carbenicillin, and coated with bacteria carrying either the *dmd-4* RNAi or the empty vector RNAi control plasmids. Worms were grown for 3 days at 20°C and then mounted on agarose slides for imaging on a compound microscope, as described in the Microscopy section below. HMC is refractory to RNAi; thus, RNAi-sensitized background strains carrying *eri-1(mg366)* (Kennedy et al., 2004) were used for these experiments.

### Temporally controlled DMD-4 protein degradation
We used conditional protein depletion with a modified auxin-inducible degradation system (Hills-Muckey et al., 2021; Negishi et al., 2021; Sepers et al., 2022). Briefly, mIAA7-tagged proteins are conditionally degraded when exposed to 5-Ph-IAA in the presence of $_{At}$TIR1$^{(F79G)}$. Strains carrying *dmd-4::linker::mIAA7::wrmScarletI3::mIAA7* [*dmd-4*(*ns1103*)] were crossed into a strain expressing $_{At}$TIR1$^{(F79G)}$ ubiquitously [*osIs182= eft-3p::AtTIR1(F79G)+LoxP+myo-2p::GFP+rps-27p::neoR+LoxP*]. Excision of *myo-2p::GFP+rps-27p::neoR* from *osIs182* was performed by injection of a plasmid driving ubiquitous Cre expression, pMDJ39 (Addgene plasmid #191381), at 20 ng/µl, and subsequent selection of transgenic F2 progeny lacking *myo-2p::GFP* expression. We dissolved 5-Ph-IAA (Cayman Chemical, 38161) in 100% ethanol to prepare a 100 mM stock solution. OP50-seeded NGM plates were coated with 5-Ph-IAA for a final concentration of 50 µM. Synchronized populations of L1 or L4 animals were transferred on 5-Ph-IAA plates and grown at 20°C for the duration of the experiment. Age-matched animals placed on OP50-seeded NGM plates coated with 100% ethanol were used as controls. The 5-Ph-IAA solutions and experimental plates were shielded from light.

### Heat-shock induced misexpression
One step RT-PCR (Invitrogen, 12594025) was used to isolate *let-381* and *dmd-4* cDNAs (primers for *let-381*: atggaatgctcaacag, ctagcaatccgataaatc; primers for *dmd-4*: atgatgatcggtaatctaca, ttatgacgattcgaatgttg), which were subsequently cloned under the heat-shock inducible promoter *hsp-16.2* by Gibson cloning (Gibson et al., 2009). These constructs were injected at 30 ng/µl to generate transgenic lines carrying *hsp-16.2::let-381* cDNA (*nsEx7466, nsEx7467*), *hsp-16.2::dmd-4* cDNA (*nsEx7468, nsEx7469*) or both (*nsEx7470, nsEx7471*) and crossed into a strain expressing GFP in HMC (*nsIs1052 [glb-26prom::gfp, myo-3p::mCherry]*). We heat shocked

25 1-day-old adult animals (P0) from each genotype by incubating parafilmed plates in a 32°C water bath for 3.5 h, followed by 2 h recovery at 20°C and a final heat shock in a 35°C water bath for 1 h. Heat shocked P0 animals were then kept at 25°C and allowed to lay progeny (F1s) for 15 h before being removed from the plate. F1 progeny were grown at 25°C for another 24 h and then scored for ectopic expression of the HMC-specific reporter. Age matched non-heat shocked animals were used as controls.

## Microscopy

Animals were anesthetized using 100 mM NaN₃ (sodium azide) and mounted on 5% agarose pads on glass slides. Z-stack images (each ~7 μm thick) were acquired using either a Zeiss confocal microscope LSM990 (images in Figs 1B,F, 2B, 3F,G, 4A, 6A,F,G; Figs S1C, S3E-I) or a Zeiss compound microscope Axio Imager M2 (images in Figs 1D, 3A,D, 5B-D; Fig. S1A) using MicroManager software (version 1.4.22) (Edelstein et al., 2010). ImageJ (Schneider et al., 2012) was used to produce maximum projections of z-stack images (2-20 slices) presented in the Figures. Figures were prepared using Adobe Illustrator.

## Quantification and statistical analysis

All microscopy fluorescence quantifications were carried out in ImageJ (Schneider et al., 2012). For each experiment, mutant (or RNAi) and control animals were imaged during the same imaging session with all acquisition parameters maintained constant between the two groups. Fluorescence intensity of gene expression in the HMC cell (Figs 3C,E, 4D-H, 5B-D) was measured in the plane with strongest signal within the z-stack in a region drawn around the HMC nucleus (for nuclear reporters) or cell body (for cytoplasmic reporters). A single circular region in an adjacent area was used to measure background intensity for each animal; this value was then subtracted from the fluorescence intensity of reporter expression for each HMC cell. Quantification of percentage of animals with reporter expression in HMC (Figs 1C, 2C, 3B, 4B, 6B,D,E; Figs S1B,D, S3A,C,D,J), quantification of percentage of animals in which HMC can be identified by DIC microscopy (Figs 1E, 4C) and quantification of percentage of HMC with morphology defects (Fig. 3H) were performed by manual counting using ImageJ. GraphPad Prism was used for graphs and statistical analysis as described in the figure legends. Unpaired two-sided Student's t-test was used to determine the statistical significance between two groups.

## Acknowledgements
We thank Oliver Hobert for strains and members of the Shaham lab for experimental advice, comments and discussion. Some strains were provided by the CGC, which is funded by NIH Office of Research Infrastructure Programs (P40 OD010440).

## Competing interests
The authors declare no competing or financial interests.

## Author contributions
Conceptualization: N.S., S.S.; Data curation: N.S.; Formal analysis: N.S., J.X., J.J.; Funding acquisition: N.S., S.S.; Investigation: N.S., J.X., J.J.; Methodology: N.S.; Project administration: S.S.; Supervision: S.S.; Validation: N.S., J.X., J.J.; Visualization: N.S.; Writing – original draft: N.S.; Writing – review & editing: N.S., S.S.

## Funding
This work was supported in part by funds from a Leon Levy Foundation Fellowship to N.S. and by National Institutes of Health grant R35NS105094 to S.S. Open Access funding provided by Rockefeller University. Deposited in PMC for immediate release.

## Data and resource availability
Strain OS15370 will be available at the CGC. All other strains and plasmids are available upon request. Any additional information required to reanalyze the data reported and requests for resources and reagents should be directed to and will be fulfilled by the corresponding author.

## Peer review history
The peer review history is available online at https://journals.biologists.com/dev/lookup/doi/10.1242/dev.204622.reviewer-comments.pdf

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
