## [Peer Review File · Development (Cambridge, England)]

***Caenorhabditis elegans* LET-381 and DMD-4 control development of the mesodermal HMC endothelial cell**

Nikolaos Stefanakis, Jasmine Xi, Jessica Jiang and Shai Shaham

DOI: 10.1242/dev.204622

Editor: Swathi Arur

Review timeline

Original submission: 26 December 2024

Editorial decision: 2 February 2025

First revision received: 16 June 2025

Accepted: 27 June 2025

Original submission

First decision letter

MS ID#: dev.204622

MS TITLE: *C. elegans* LET-381/FoxF and DMD-4/DMRT control development of the mesodermal HMC endothelial cell

AUTHORS: Nikolaos Stefanakis, Jasmine Xi, Jessica Jiang and Shai Shaham

Dear Dr Shaham,

I have now received all the referees' reports on the above manuscript, and have reached a decision. The referees' comments are appended below, or you can access them online: please go to:

As you will see, the referees express considerable interest in your work, and make recommendations to further improve the clarity and the rigor of the work. If you are able to revise the manuscript along the lines suggested, which may involve further experiments, I will be happy receive a revised version of the manuscript. Your revised paper will be re-reviewed by one or more of the original referees, and acceptance of your manuscript will depend on your addressing satisfactorily the reviewers' major concerns. Please also note that Development will normally permit only one round of major revision.

Please attend to all of the reviewers' comments and ensure that you clearly highlight all changes made in the revised manuscript. Please avoid using 'Tracked changes' in Word files as these are lost in PDF conversion. I should be grateful if you would also provide a point-by-point response detailing how you have dealt with the points raised by the reviewers in the 'Response to Reviewers' box. If you do not agree with any of their criticisms or suggestions please explain clearly why this is so.

Reviewer 1

Advance summary and potential significance to field

This manuscript from Stefanakis et al explores the role of let-381, the *C. elegans* homolog of the forkhead transcription factor FoxF, in fate specification of the HMC cell. The HMC cell is derived from the mesoderm and is put forth here as a model for understanding the specification of mesodermal cells (e.g. endothelial cells) in vertebrates.

The data presented is very straightforward and clear cut, building on a previous publication to establish a gene regulatory network of three transcription factors that specify and maintain the HMC. Strengths of the data are that multiple promoter lines are analyzed, and that reciprocal experiments on *dmd-4* and *let-381* are performed. A GRN is presented that summarizes the data. However, on its own, the fact that *let-381* specifies cell fate isn't necessarily novel, since forkhead transcription factors are known to have this role. If the novelty is that this GRN is operating in a non-contractile cell, the connection between the HMC and other systems needs to be strengthened. This could be achieved by some of the experiments listed below, which are presented as suggestions and not necessarily required.

Comments for the author

Major comments:

As it is presented, there is insufficient information describing the ns1023 and ns1026 alleles, and it's puzzling (or interesting) that the insertion of these random 34bp restores expression. How does this work?

In Figure 3, several promoter fusions are analyzed. Is there any biological significance to these different promoters in worms? This should be explained more clearly (e.g. more information than '2 endogenous and 4 transgenic reporters'). Are they connected to endothelial biology in any way?

Figure 3F shows clearly that there is a morphological difference, but this difference is all lumped into a 'defective' category in 3G. Do 100% of the HMC cells show this morphology?

The difference in HMC morphology implies that a cell fate switch has occurred and is further supported by data in Figure 4I/J, but additional data could support this claim. For example: If *dmd-4* is depleted with an AID, how long does it take for *nep-2* expression to appear? Do morphological or functional changes accompany this defect in maintenance?

It isn't clear to me what is gleaned from the promoter analysis in Figure 4B, since according to DIC the HMC is gone, and the promoters lack the *dmd-4* binding sites.

How identification of this GRN applies to other systems could be elaborated further in the discussion. For example: are any downstream targets known? Is a *dmd-4* homolog expressed in these other cell types?

Minor comments:

Planaria is the common name of these flatworms; it shouldn't be italicized or capitalized.

Reviewer 2

Advance summary and potential significance to field

In this very nice paper, Stefanakis et al. describe the differentiation program of an interesting and previously quite mysterious cell in the nematode *C. elegans*. The cell, called *hmc*, has recently been recognized as being essentially the worm homolog of endothelial cells, thereby providing some interesting overall context/importance to study its differentiation program. The authors define two transcription factors that specify the fate of this neuron, using rigorous, well described and well-documented genetic approaches. This manuscript is of significant interest to the readership of *Development* and I only have only minor requests/suggestions for improvement:

Experimental:

- 1) The authors base their claim on *dmd-4* being "downstream" of *let-381* (a point they find important enough to mention in the abstract) on *dmd-4(RNAi)*, in which they observe no effect on *let-381* expression. RNAi often does not generate null mutant phenotype and since the authors have done other *dmd-4* mutant analysis with the null allele, they should do the same with their *let-381* reporter. If this result holds up with the null, it may be useful to use somewhere in the manuscript

(Discussion?) the term "feedforward loop" for this regulatory architecture, as seen also for *let-381* in the *cc*'s (and the classic example of course being *unc-86* and *mec-3*).

2) This comment may appear a little nit-picky, but I'm wondering whether the authors can infer a maintenance role of *let-381* with their autoregulatory allele if the only thing they look at is adult animals. Meaning, in this autoregulatory allele, the initiation of differentiation may already be affected. If the authors could look for proper expression of 1 or 2 of these markers in embryo and/or L1 and see LESS of an effect than they see in the adult, this issue would be settled.

3) The authors should describe the *hmc* morphology defect at the very least with a few more words (incl. what their scoring criteria are). Ideally, the authors would show a nice 3d rendering of wt *hmc* with a confocal and a few examples of defects. I'll leave this to the authors.

4) Also optional: I'm wondering whether the Emb phenotypes of *dmd-4* and *let-381* are related at all, i.e. whether they may collaborate in other cell(s), too. There's no definite proof for why *dmd-4* dies - but it's so restrictively expressed that *pm8* and/or the VPI are the best candidates. Any indication that *let-381* is expressed in these cells? If so, it may be worth raising this point in the Discussion.

Editorial changes:

p.3: Please change "transcriptional programs involved in endothelial cell differentiation and subtype specification and maintenance remain largely unknown." to something like "incompletely understood" -> such softer wording is necessitated by refs such as PMID 23620236, 36713285 (which should be cited).

p.4: please add Altun & Hall WormAtlas citation to the two Choi et al. 2013 refs.

p.6: please add some speculation why the two regulatory alleles of *let-381* behave so differently, given that they both affect the *let-381* motif. I know, it's impossible to know for sure what's happening, but it would be nice to see it at least acknowledged that in one cell *let-381* requires its own binding site for autoregulation, while in the other case it doesn't.

p.10: please add Altun & Hall WormAtlas citation to Fares and Greenwald Ref

p.10: *unc-86* is perhaps the absolute front-runner in terms of a TF that's known to work on conjunction with other TFs in distinct cellular context (both historically, as well as numerically, as well as in different organisms). Please consider adding Ref PMID 32012462

p.11: Yes, terminal selectors repress alternative fates; the authors could perhaps cite another example: PMID: 36178933 (in addition to Feng and Remesal)

Reviewer 3

Advance summary and potential significance to field

In this manuscript, the authors studied the mechanisms underlying the specification and maintenance of the *C. elegans* Head Mesodermal Cell (HMC) fate. The authors found that the Forkhead transcription factor LET-381/FoxF is expressed in the HMC and functions cell autonomously to specify the HMC fate and maintain HMC gene expression. Furthermore, the Dmrt transcription factor DMD-4 functions downstream of and works together with LET-381 in this process. They further showed that DMD-4 also represses the expression of a gene normally expressed in GLR glia, a different mesodermal cell type. Because the HMC shares some similarity to vertebrate endothelial cells, the finding from this work is potentially informative in our understanding of how endothelial cells may be specified.

Comments for the author

While the study is solid and the conclusions are consistent with LET-381 and DMD-4 playing roles to specify the HMC fate, what is lacking in this study is evidence showing that ectopic expression of LET-381 and DMD-4 in other lineages is sufficient to specify HMC fate. Furthermore, there is no evidence supporting DMD-4 being the factor cooperating with or working together with LET-381 for HMC specification. Do any of the HMC genes appear to be directly regulated by LET-381 and/or DMD-4? Are there putative binding sites for LET-381 and/or DMD-4 in the promoters of HMC genes? In addition, it is peculiar that only one gene that is normally expressed in GLR glia is inappropriately expressed in the HMC upon *dmd-4* RNAi.

Minor:

Figure 3A, the third panel is incorrectly labeled. It should be *let-381(ns1026)*, not *ns1023*.

Figure 3 legend: throughout the legend for this figure, *glb-26promoter::gfp* and *dmd-4promoter::mCherry* and names of other reporters should be italic.

First revision

Author response to reviewers' comments

We would like to thank the referees for taking the time to review our manuscript and for their constructive comments. We have now addressed each concern raised in the review with new experiments and text changes. Below is a point-by-point response to referee comments:

Reviewer 1

Major comments:

1. As it is presented, there is insufficient information describing the *ns1023* and *ns1026* alleles, and it's puzzling (or interesting) that the insertion of these random 34bp restores expression. How does this work?

As described in the materials and methods, we used CRISPR/Cas9 to generate a 106 bp deletion of the *let-381* binding site in the *let-381* promoter region (allele *ns1023*). From this experiment, we also isolated an indel allele, *ns1026*, in which the 106 bp region is replaced by a 34 bp sequence of unknown origin, presumably an artifact of the double strand break repair. These two alleles behave differently with respect to *let-381* expression in the HMC cell. At the reviewer's suggestion, we have now clarified the molecular details of each lesion and its effects on *let-381::gfp* expression in Figures 2A and 2B and in the materials and methods.

We agree with the reviewer that the difference between the *ns1023* and *ns1026* alleles is intriguing. Although deciphering the mechanistic difference between these alleles is beyond the scope of the current manuscript, as we are using these alleles as tools and their mechanism of action is secondary to their utility for our studies, we speculate that the 34 bp insertion sequence contains a motif recognized by a transcription factor expressed in HMC, and not in GLR, that can now drive *let-381* expression in HMC. We have now added discussion of this possibility on page 7 of the revised manuscript.

2. In Figure 3, several promoter fusions are analyzed. Is there any biological significance to these different promoters in worms? This should be explained more clearly (e.g. more information than '2 endogenous and 4 transgenic reporters'). Are they connected to endothelial biology in any way?

While *gbb-2/GABBR2* (metabotropic GABA receptor) and *snf-11/SLC6A1* (GABA transporter) are both involved in GABA signaling, in general, the genes shown in Figure 3B are not functionally related, and they were chosen only because they are known HMC markers. This is now explained on page 7 of the revised manuscript.

Regarding a connection to endothelial cell biology, it is noteworthy that orthologs of some of the genes shown in Figure 3 (*gly-18/GCNT2* glucosaminyl transferase, *gbb-2/GABBR2* Gaba receptor, *p11-1/PLCL1* phospholipase C) are enriched in endothelial cells in the mammalian brain. We now mention this, providing supporting references, in the Discussion section (page 14 of the revised manuscript).

Also related to Figure 3B, in the original version of the manuscript, the *glb-26prom::gfp* data was from 3 extrachromosomal array strains. For greater rigor, we have now replaced this data with scoring of a new genomically integrated *gfp* fusion to the endogenous *glb-26* locus we generated, with identical results.

3. Figure 3F shows clearly that there is a morphological difference, but this difference is all lumped into a 'defective' category in 3G. Do 100% of the HMC cells show this morphology?

We thank the reviewer for this comment. We now provide a more detailed description of the morphological differences, with additional examples in Figure 3G and in Supplementary Movies S1 - 4 (of the revised manuscript).

4. The difference in HMC morphology implies that a cell fate switch has occurred and is further supported by data in Figure 4I/J, but additional data could support this claim. For example: If *dmd-4* is depleted with an AID, how long does it take for *nep-2* expression to appear? Do **morphological or functional** changes accompany this defect in maintenance?

We thank the reviewer for this important comment and agree that understanding the correlation in timing between gene expression changes and morphological changes is important. Motivated by the reviewer comment, we used CRISPR/Cas12 to generate an endogenously tagged *dmd-4::mScarlet13::aid* allele and found that *nep-2p7::gfp* expression in HMC appears after ~16 hours on auxin for L1 animals and after ~6 hours on auxin for L4 animals. HMC cells with ectopic *nep-2* expression initially appear wild type in morphology but start showing morphological defects ~24 hours after initial auxin exposure, supporting the idea that gene expression changes may drive morphological changes. This is described in the Results section (page 11), Fig. 6, and Fig. S3 of the revised manuscript.

5. It isn't clear to me what is gleaned from the promoter analysis in Figure 4B, since according to DIC the HMC is gone, and the promoters lack the *dmd-4* binding sites.

Although by DIC the HMC cannot be identified in *dmd-4* mutants, it remains possible that the cell is still there, but exhibits altered morphology. In this case, the altered cell may still express some HMC genes. We, therefore, believe that testing many known HMC reporters and showing that none of them is expressed in the HMC in the *dmd-4(ot933)* null mutant, strengthens our argument that the HMC is not specified in the absence of DMD-4. Also, we now show that *p11-1* has a functional *dmd-4* motif in its promoter (revised Fig 5D) and it is possible that some of the other genes could also contain functional *dmd-4* binding sites.

6. How identification of this GRN applies to other systems could be elaborated further in the discussion. For example: are any downstream targets known? Is a *dmd-4* homolog expressed in these other cell types?

We thank the reviewer for this comment and have added additional discussion here on page 14 of the revised manuscript. Briefly, although targets of *Foxf* proteins in mesodermal endothelial and mural cells in the mammalian brain are not well studied, save a report that TGF β signaling is downregulated in *Foxf2* mutants, we found that homologs of some *C. elegans let-381* targets are enriched in mammalian endothelia (see response to comment #2 above). With respect to *dmd-4/Dmrt*, recent scRNA-seq studies show expression of *Dmrt2* in brain endothelial cells, suggesting possible conservation with *C. elegans*.

Minor comments:

Planaria is the common name of these flatworms; it shouldn't be italicized or capitalized.

Thank you for noticing this. We have now fixed this throughout the revised manuscript.

Reviewer 2

1) The authors base their claim on *dmd-4* being "downstream" of *let-381* (a point they find important enough to mention in the abstract) on *dmd-4*(RNAi), in which they observe no effect on *let-381* expression. RNAi often does not generate null mutant phenotype and since the authors have done other *dmd-4* mutant analysis with the null allele, they should do the same with their *let-381* reporter. If this result holds up with the null, it may be useful to use somewhere in the manuscript (Discussion?) the term "feedforward loop" for this regulatory architecture, as seen also for *let-381* in the cc's (and the classic example of course being *unc-86* and *mec-3*).

We thank the reviewer for this suggestion. As shown in Figure 4B, *let-381:gfp* expression is not observed in the *dmd-4(ot933)* null mutant, a result different from the *dmd-4* RNAi. However, because we believe that the HMC cell is not properly specified in *dmd-4(ot933)* mutants, we cannot use this result to infer any genetic interaction between the two transcription factors, at least in early steps of HMC specification. It is indeed possible that early on *let-381* and *dmd-4* act in parallel. As shown in our suggested model, Figure 6H, we believe that *dmd-4* acts downstream of *let-381* in fate maintenance. The two experiments supporting this are: (a) *dmd-4* expression is reduced in *let-381* postembryonic knockdown (Fig. 3C) and *let-381* expression is unaffected in *dmd-4* RNAi (Fig. 4H). We have now clarified this in the abstract and text (page 9).

2) This comment may appear a little nit-picky, but I'm wondering whether the authors can infer a maintenance role of *let-381* with their autoregulatory allele if they only thing they look is are adult animals. Meaning, in this autoregulatory allele, the initiation of differentiation may already be affected. If the authors could look for proper expression of 1 or 2 of these markers in embryo and/or L1 and see LESS of an effect than they see in the adult, this issue would be settled.

We thank the reviewer for this suggestion to strengthen the argument that *let-381(ns1023)* affects maintenance of HMC gene expression. We now provide images and quantification of HMC expression of two reporters (*hot-5p::gfp* and *glb-26::gfp*) at the L1 larval stage. For both reporters, we observed that GFP expression is seen in some HMC cells in L1 *let-381(ns1023)* animals, although at levels lower than wild type. By contrast, GFP expression is not detected in HMC cells of L4 stage *let-381(ns1023)* animals. This suggests that initiation of expression in the HMC is not affected. We now explain this in the manuscript (page 7) and in relevant images (Fig. S1A-D).

3) The authors should describe the hmc morphology defect at the very least with a few more words (incl. what their scoring criteria are). Ideally, the authors would show a nice 3d rendering of wt hmc with a confocal and a few examples of defects. I'll leave this to the authors.

We agree with the reviewer, and we have now provided a better description of the HMC morphology defects and additional images/movies showing more examples (also see response to comment #3 from Reviewer 1 above).

4) Also optional: I'm wondering whether the Emb phenotypes of *dmd-4* and *let-381* are related at all, i.e. whether they may collaborate in other cell(s), too. There's no definite proof for why *dmd-4* dies - but it's so restrictively expressed that pm8 and/or the VPI are the best candidates. Any indication that *let-381* is expressed in these cells? If so, it may be worth raising this point in the Discussion.

This is an interesting point. GFP of endogenously tagged *let-381::gfp* is detected only in GLR glia, HMC cell and coelomocytes in postmitotic stages. We did not detect any *let-381::gfp* expression in the pharynx. It is possible that LET-381 could be expressed in pm8 and VPI precursor cells, although such expression is not observed in published embryonic lineaging experiments using a LET-381 protein fusion (EPIC Expression patterns in *C. elegans*). Based on our own unpublished observations and the information from the *wormatlas.org* Chapter on GLR glia, GLR-ablation causes worms to arrest at the L1 stage, suggesting that *let-381*-related embryonic lethality may be, at least in part, due to GLR glia specification defects. By contrast, DMD-4 is not expressed in the GLR glia. Therefore, if *dmd-4*-related lethality is indeed due to pm8 and VPI defects, we would then expect that lethality of the two genes is not related. Alternatively, embryonic lethality of both genes could

be due to specification defects of the HMC cell, since this is the cell where expression of both genes overlaps. HMC-cell ablations could address this possibility.

Because there is good reason to believe that the lethality occurs for different reasons in *dmd-4* and *let-381* mutants, we opted not discuss this issue in the manuscript, to avoid distraction from the main points we wish to convey.

Editorial changes:

p.3: Please change "transcriptional programs involved in endothelial cell differentiation and subtype specification and maintenance remain largely unknown." to something like "incompletely understood" -> such softer wording is necessitated by refs such as PMID 23620236, 36713285 (which should be cited).

We thank the reviewer for pointing out this literature. These references are now added and phrasing changed.

p.4: please add Altun & Hall WormAtlas citation to the two Choi et al. 2013 refs.

Done

p.6: please add some speculation why the two regulatory alleles of *let-381* behave so differently, given that they both affect the *let-381* motif. I know, it's impossible to know for sure what's happening, but it would be nice to see it at least acknowledged that in one cell *let-381* requires its own binding site for autoregulation, while in the other case it doesn't.

We thank the reviewer for this comment; we have now added a possible explanation in the manuscript (see also our response to comment #1 from Reviewer 1).

p.10: please add Altun & Hall WormAtlas citation to Fares and Greenwald Ref

Done

p.10: *unc-86* is perhaps the absolute front-runner in terms of a TF that's known to work on conjunction with other TFs in distinct cellular context (both historically, as well as numerically, as well as in different

Done

p.11: Yes, terminal selectors repress alternative fates; the authors could perhaps cite another example: PMID: 36178933 (in addition to Feng and Remesal)

Done

Reviewer 3

1. While the study is solid and the conclusions are consistent with LET-381 and DMD-4 playing roles to specify the HMC fate, what is lacking in this study is evidence showing that ectopic expression of LET-381 and DMD-4 in other lineages is sufficient to specify HMC fate.

We thank the reviewer for this suggestion. To address this issue, we pursued a strategy similar to what we had done previously, showing that LET-381 and UNC-30 overexpression yields ectopic GLR glia gene expression (Stefanakis et al., 2024). Specifically, we cloned *let-381* cDNA and *dmd-4* cDNA under a heat-shock inducible promoter. Heat shock of either *let-381*, *dmd-4* or a combination of both cDNAs was not sufficient to induce ectopic expression of the HMC marker we tested. This is now described in the revised manuscript (page 10) and in Fig. S2A.

2. Furthermore, there is no evidence supporting DMD-4 being the factor cooperating with or working together with LET-381 for HMC specification. Do any of the HMC genes appear to be directly regulated by LET-381 and/or DMD-4? Are there putative binding sites for LET-381 and/or DMD-4 in the promoters of HMC genes?

This is an important question. To address it, we manually searched for LET-381 and DMD-4 motifs in the regulatory regions of the HMC genes in Figure 3B. We found that almost all these genes contain such motifs. Using CRISPR/Cas9, we mutated *let-381* motifs in *pll-1* and *gbb-2* and *dmd-4* motifs in *pll-1*. Mutation of the motifs results in significant reduction of expression of the downstream gene, suggesting that LET-381 and DMD-4 together may be direct regulators of HMC gene expression. These experiments are now described in the revised manuscript (page 9) and in Fig. 5.

In addition, it is peculiar that only one gene that is normally expressed in GLR glia is inappropriately expressed in the HMC upon *dmd-4* RNAi.

One possibility to explain why only one of our reporters was affected by *dmd-4* knockdown is that DMD-4 only represses some but not all GLR genes in HMC. Alternatively, *dmd-4* RNAi may only weakly reduce DMD-4 function. To experimentally address this issue, we knocked down DMD-4 function using auxin inducible degradation. In this setting, we were able to detect ectopic expression of an additional GLR-reporter, *lgc-55prom::gfp*, which did not appear derepressed by *dmd-4*(RNAi). These experiments are now discussed in the manuscript (page 11) and Fig. S3H-J. Unfortunately, we could not test the other two GLR reporters (*unc-30::gfp::aid* and *hlh-1::gfp::aid*) with DMD-4:AID as they are already tagged with AID and their expression is abolished in the presence of auxin.

Minor:

Figure 3A, the third panel is incorrectly labeled. It should be *let-381(ns1026)*, not *ns1023*.

Thank you for noticing this. We have now fixed it.

Figure 3 legend: throughout the legend for this figure, *glb-26promoter::gfp* and *dmd-4promoter::mCherry* and names of other reporters should be italic.

Thank you for noticing this. We have now fixed it.

Second decision letter

MS ID#: dev.204622R1

MS TITLE: *C. elegans* LET-381/FoxF and DMD-4/DMRT control development of the mesodermal HMC endothelial cell

AUTHORS: Nikolaos Stefanakis, Jasmine Xi, Jessica Jiang and Shai Shaham

Dear Dr Shaham,

I am happy to tell you that your manuscript has been accepted for publication in Development, pending our standard publication integrity checks.

Reviewer 1

Advance summary and potential significance to field

The authors have addressed all of my concerns. The new data added to Fig. 6 strongly support the role of *dmd-4* in maintenance of morphology.

Reviewer 2

Advance summary and potential significance to field

The authors have done a terrific job in responding to my comments (and the other reviewer's comments, I believe) and the paper is ready to be published as is.

Reviewer 3

Advance summary and potential significance to field

This revised manuscript has addressed previous reviewers' comments very well, and it provides compelling evidence supporting the transcription factors LET-381/FoxF and DMD-4/DMRT functioning together to control the development of the head mesodermal cell in *C. elegans*. The work will be of interest to readers of *Development*.